Floral pathway integrator gene expression mediates gradual transmission of environmental and endogenous cues to flowering time

van Dijk Aalt D.J. aaltjan.vandijk@wur.nl 1 2 3
Molenaar Jaap 1
1 Biometris, Department for Mathematical and Statistical Methods, Wageningen University , Wageningen , The Netherlands
2 Laboratory of Bioinformatics, Wageningen University , Wageningen , The Netherlands
3 Bioscience, Wageningen University and Research , Wageningen , The Netherlands
Beemster Gerrit
Electronic publication date: 2017 Apr 19
Publication date: 2017
Volume: 5
Electronic Location ID: e3197
Received 2017 Jan 11; Accepted 2017 Mar 17
Copyright: ©2017 van Dijk and Molenaar
Copyright year: 2017
Copyright holder: van Dijk and Molenaar
License: This is an open access article distributed under the terms of the Creative Commons Attribution License, which permits unrestricted use, distribution, reproduction and adaptation in any medium and for any purpose provided that it is properly attributed. For attribution, the original author(s), title, publication source (PeerJ) and either DOI or URL of the article must be cited.
License URL: https://creativecommons.org/licenses/by/4.0/

Keywords: Arabidopsis thaliana, Flowering time, Linear regression, Gene expression levels

Funding: The authors received no funding for this work.

==============================
The appropriate timing of flowering is crucial for the reproductive success of plants. Hence, intricate genetic networks integrate various environmental and endogenous cues such as temperature or hormonal statues. These signals integrate into a network of floral pathway integrator genes. At a quantitative level, it is currently unclear how the impact of genetic variation in signaling pathways on flowering time is mediated by floral pathway integrator genes. Here, using datasets available from literature, we connect Arabidopsis thaliana flowering time in genetic backgrounds varying in upstream signalling components with the expression levels of floral pathway integrator genes in these genetic backgrounds. Our modelling results indicate that flowering time depends in a quite linear way on expression levels of floral pathway integrator genes. This gradual, proportional response of flowering time to upstream changes enables a gradual adaptation to changing environmental factors such as temperature and light.

Introduction

The reproductive success of flowering plants depends on flowering at the right moment. Hence, plants have evolved genetic and molecular networks integrating various environmental cues with endogenous signals in order to flower under optimal conditions (Srikanth & Schmid, 2011). The signal transduction pathways that receive and transmit input signals include the photoperiod pathway, the vernalization pathway, the ambient temperature pathway, and the autonomous pathway (Andres & Coupland, 2012). The input from these pathways is integrated by a core set of floral pathway integrator genes (Simpson & Dean, 2002). The regulation of flowering time by these various factors has been extensively studied experimentally in the plant model species Arabidopsis thaliana. Substantial qualitative information is available about the factors involved and how these interact genetically, both for the signal transduction pathways and the floral pathway integrator genes (Bouche et al., 2016). Activation of the photoperiodic flowering pathway leads to transcriptional activation of FLOWERING LOCUS T (FT), an activator of flowering. FT is produced in the leaves and moves to the shoot apical meristem (Zuo et al., 2011), leading to activation of SUPPRESSOR OF OVEREXPRESSION OF CONSTANS 1 (SOC1) (Yoo et al., 2005) and APETALA1 (AP1) expression (Abe et al., 2005; Wigge et al., 2005). The vernalization (winter cold) pathway inhibits the transcription of FLOWERING LOCUS C (FLC). FLC, together with SHORT VEGETATIVE PHASE (SVP), represses the transcription of SOC1 and FT. Thus FLC acts as a flowering repressor by blocking the photoperiodic flowering pathway. In the ambient temperature pathway, which involves amongst other FLOWERING LOCUS M (FLM) and SVP, small fluctuations in temperature influence flowering time via floral pathway integrators including FT and SOC1 (Verhage, Angenent & Immink, 2014; Capovilla, Schmid & Pose, 2015). SOC1 integrates signals from multiple pathways and transmits the outcome to LEAFY (LFY) (Immink et al., 2012; Michaels et al., 2003); SOC1 is supposed to act at least partially via a positive feed-back loop in which AGAMOUS-LIKE 24 (AGL24) is involved upon dimerizing with SOC1 (Lee et al., 2008). Autonomous pathway mutants are characterized by delayed flowering irrespective of day length. The proteins encoded by the genes in the autonomous pathway generally fall into two broad functional categories: general chromatin remodelling or maintenance factors, and proteins that affect RNA processing (Srikanth & Schmid, 2011). Gibberellins influence the floral transition through the regulation of SOC1 and LFY (Eriksson et al., 2006). LFY is a positive regulator of AP1 (Wagner, Sablowski & Meyerowitz, 1999) and the commitment to flower is ascertained by a direct positive feed-back interaction between AP1 and LFY. Once the expression of AP1 has been initiated, this transcription factor orchestrates the floral transition by specifying floral meristem identity and regulating the expression of genes involved in flower development (Kaufmann et al., 2010).

In addition to qualitative information on pathways involved in flowering time regulation, recently quantitative information has become available. This includes flowering time measurements under various conditions and in different genetic backgrounds (Jung & Muller, 2009; Leal Valentim et al., 2015), and time series of expression for key floral pathway integrator genes (Leal Valentim et al., 2015). Such quantitative information has enabled construction of a set of models describing flowering time regulation at the molecular level (Leal Valentim et al., 2015; Jaeger et al., 2013; Dong et al., 2012; Salazar et al., 2009). Given the above-described complexity, computational models are useful tools to comprehend flowering time regulation. One example of a quantitative finding from our model (Leal Valentim et al., 2015) for the network of floral pathway integrator genes is that a disturbance in a particular gene has not necessarily the largest impact on directly connected genes. For example, the model predicts that SOC1 mutation has a larger impact on AP1, which is not directly regulated by SOC1, compared to its effect on LFY which is under direct control of SOC1. This prediction was confirmed by expression data.

Flowering time regulation facilitates the successful dispersion of flowering plants over the world (Andres & Coupland, 2012) by contributing to the adaptation of plants to different environmental conditions. In this context, it is an important question how genetic variation in the various signaling pathways influences flowering time regulation. Can we describe the effect of genetic variation in these signaling pathways by linking the magnitude of flowering time change to the magnitude of expression change of floral pathway integrator genes? If so, what type of relationship exists between expression levels of floral pathway integrator genes and flowering time in genetic backgrounds which differ in signaling components?

The above-mentioned quantitative analyses focus on one specific Arabidopsis genetic background, without genetic difference in signaling pathways being taken into account, leaving these questions so far unanswered. In principle, one could imagine answering these questions by extending these models to include a large number of signaling pathway components. However, construction of such large models would lead to serious complications in terms of e.g., parameter estimation. Here we follow a different route to investigate how the effect of genetic variation in components of upstream signalling pathways on flowering time is mediated by floral pathway integrator genes. We establish a quantitative connection between expression levels of floral pathway integrator genes, and flowering times in various genetic backgrounds differing in upstream signal components. This demonstrates that in many cases, floral pathway integrator genes transmit perturbations to flowering time via gradual, proportional changes in their expression levels. Our current study is complementary to our previous modelling approach which focused on the floral pathway integrator gene network, and not on the input to this network by upstream signalling components. This analysis provides a quantitative understanding of the effect of variation in the various input pathways on flowering time, which will ultimately enable us to better understand plant adaptation.

Methods

Simulations

Predictions from the dynamic flowering time model were obtained using the model as presented in Leal Valentim et al. (2015). This consists of a set of Ordinary Differential Equations (ODEs) for the dynamics of AP1, LFY, SOC1, FD, FT and AGL24; SVP and FLC are present as external inputs in the model. In each of the six ODEs, regulation of gene expression is described by one or more terms of the form β∗f(x), where f is a function of concentrations x of one or more regulators. To simulate the effect of genetic variation in upstream signalling pathways influencing a given gene, the value of each parameter β in its equation was modified by multiplying it with a factor a ranging from 0.05 to 10 in steps of 0.05 and subsequently from 10 to 100 in steps of 1. The resulting flowering time after simulating the modified model was obtained, as well as the expression value of the gene itself at day 10 (this timepoint was used because it matches closely with the timepoint used in much of the experimental datasets that we used). Out of the resulting expression values, a range of ten-fold expression change was chosen around the unperturbed expression level at day 10. In addition, in Fig. 1 a five-fold expression range around the unperturbed expression level at day 10 is indicated. These ten-fold and five-fold ranges were obtained by dividing or multiplying the unperturbed value at day 10 with sqrt(10) = 3.16 or sqrt(5) = 2.236, respectively. For SVP and FLC there is no ODE because these genes are present as external inputs in the model. For these, variation in upstream signalling pathways was simulated by simply setting the level of the gene to different fixed levels. For SVP this again involved a range of ten-fold expression change; for FLC this range was arbitrarily made larger because of the small effect of ten-fold expression change.

Figure 1 Dynamic model predicts linear dependency of flowering time in different genetic backgrounds on floral pathway integrator gene expression levels.

The dynamic Ordinary Differential Equation (ODE) model for flowering time regulation in Leal Valentim et al. (2015) was used to simulate how flowering time (FLT) depends on gene expression level measured at day 10 for (A) AGL24 (B) SOC1 (C) LFY (D) FT (E) SVP (F) FLC. To mimic genetic variation in upstream signalling pathways, parameter values in the ODE model were modified as explained in Methods. Red points indicate the expression level of the gene at day 9–11 in the unperturbed model. Vertical dotted grey lines indicate five-fold expression range around the expression level at day 10 in the unperturbed model, which is indicated with a vertical dotted red line. For FLC, the five-fold range is small compared to the displayed range and the vertical lines fall on top of each other.

Experimental data

We use data from a randomly chosen subset of genes for which mutations are described as impacting flowering time (Lloyd & Meinke, 2012). Our dataset has at least several examples per floral pathway integrator gene. Data was extracted from figures or tables in papers describing the effect of mutations of particular genes on flowering time, and presenting the expression level of genes involved in signal integration. Expression measurements in different experiments are made at different days and/or different tissues, but such differences are not taken into account. Also, in particular for FT, often values are provided for several timepoints during one day (to capture the circadian rhythm). Although for such a case in principle it would be best to record the total area under the curve (sum of expression), for simplicity the highest observed value was used as approximation in this case.

To analyse the data, a straight line was fitted through each of the datasets: T = Sensitivity∗x + T0, where T is flowering time and x is expression level; Sensitivity and T0 are parameters for which values are obtained in the fit. The R-function lm was used for the linear fit, and cor.test to test the statistical significance.

One important point in our data analysis is that various datasets were obtained using different ways of normalizing the expression values. Multiplicative normalization should effect Sensitivity in a multiplicative way: if T = S∗x + T0, then for x′ = a∗x, T = (S∕a)∗x′ + T0, i.e., S′ = S∕a. Hence, we can compare the value of Sensitivity for different genes only when the same reference gene is used for normalization, and no additional relative normalization is used. The parameter T0 should be independent of the normalization that is used for expression data. It would only depend on the unit of flowering time. This unit was either total leaf number or rosette leaf number; we did not observe a systematic difference for data reported in either unit and hence did not discriminate between these cases in presenting our results.

In addition to separately fitting the various datasets available for a given floral pathway integrator gene, we also obtained one model for each floral pathway integrator gene in which the various datasets were fitted simultaneously. This was performed using the R-function nls. In these models, each dataset obtained its own value of Sensitivity, but only one global value of T0 was used for each floral pathway integrator gene.

Results

We aim to obtain a comprehensive picture of how variation in signalling pathways influences flowering time via affecting floral pathway integrator genes. To do so, we first analysed our recently published mechanistic model for the floral pathway integrator gene network (Leal Valentim et al., 2015). This model describes regulatory interactions between the various integrator genes and is able to predict the effect of a specific perturbation in one of the genes, on all the other genes in the network. By assessing how this finally influences AP1 expression, the model predicts flowering time: flowering is predicted to start when AP1 expression passes a certain threshold. This model was developed using expression data and flowering time of wild-type Arabidopsis thaliana, as well as mutants of floral pathway integrator genes. In our current work, we focus on genetic variation in upstream signalling pathways, which were not used previously for modelling. To simulate variation in these upstream signalling pathways, parameters describing input to the floral pathway integrator genes were modified in the model (see ‘Methods’). This allowed to observe the dependency of predicted flowering time on expression levels of floral pathway integrator genes (Fig. 1). These plots indicate that for each gene, in an expression range of five- to tenfold around its nominal expression, the response of flowering time to expression change is approximately linear. To further analyse the response curves obtained from our model (Fig. 1) a linear model was fitted. The p-value associated with the linear fit is significant (<10−15) for all the genes over the full range of expression displayed in Fig. 1. The obtained Pearson R2 values for the linear fits are all above 0.75.

Hence, analysis of our floral pathway integrator gene regulatory network model predicts a gradual and rather linear dependence of flowering time response on changes in input to the floral regulatory network. To assess the validity of this prediction, we chose to analyze large amounts of datasets available in literature. Numerous studies present measurements of flowering times in various conditions and for various genetic backgrounds. Since one often knows which floral pathway integrator gene is relevant for the specific signalling pathway involved, the expression levels of the specific gene thought to be responsible for mitigating the input from the signal transduction pathway are measured as well. Although one has to extract most of this data manually from tables or figures in relevant publications, it is an advantage that large amounts of data can be analysed in this way. Even though some of the individual datasets are small, in its totality the data consists of over 200 pairs of measurements of expression level and flowering time. This data has so far been scattered throughout literature and we demonstrate that it can be integrated. We use this data as a means to describe in a quantitative way the effect of changes in genetic background in signalling pathway components on flowering time. We start with a specific example regarding the floral pathway integrator gene SOC1.

Introductory example for SOC1

SOC1 expression measurements (qPCR) were obtained in different genetic backgrounds (cry2 and fri, affecting the photoperiod pathway and the vernalization pathway, respectively) and different conditions (El-Din El-Assal et al., 2003). For the same conditions, flowering time was also measured (El-Din El-Assal et al., 2003). It is straightforward to combine these two sets of measurements in a quantitative way, although this has not yet been done so far. As shown in Fig. 2A, across the different genetic backgrounds, there is a quite strong linear dependency of flowering time on the expression level of SOC1 (R2 = 0.80). It is this dependency that is the focus of investigation of this study, for SOC1 as well as for floral pathway integrator genes. In our analysis, we focus on the effect of differences in genetic backgrounds on each particular gene in the floral pathway integrator gene network. For that particular gene, expression level measurements might then be explanatory for flowering time changes. By analysing data as shown in Fig. 2A from various publications, we are able to get a comprehensive quantitative picture how floral pathway integrator gene expression mediates transmission of environmental and endogenous cues to flowering time.

Figure 2 Dependency of flowering time (vertical axis) on SOC1 expression levels (horizontal axis) in various genetic backgrounds and various conditions, obtained in three different studies (A–C).

Flowering time is reported in number of leaves; expression is normalized by scaling to wildtype expression level (A), normalized to actin (B) or normalized to tubulin (C).

When integrating and comparing data for different experiments or different genes, one particular complication is that reported qPCR gene expression levels are normalized in various ways. In order to be able to combine datasets from different publications, one of the two following conditions should hold: (1) The same reference gene was used for normalization, and we assume that the expression level of the reference gene is constant in the different conditions applied in the various publications. In this scenario, expression levels of different genes in various publications can be quantitatively compared. Alternatively, (2) the reported expression level was scaled using wildtype expression levels of the gene of interest. In this case, in order to compare data from different publications, it is essential that the wildtype expression level that is used is the same. This seems less likely than the assumption that a reference gene such as actin or tubulin has a constant gene expression level. In several cases, the two scenarios are combined, in the sense that qPCR data are first normalized to a reference gene but that the reported expression level is subsequently scaled to a wildtype expression level.

For SOC1, the data analysed above were reported after scaling the expression level to wildtype SOC1 expression levels. Two additional examples of data for flowering time and SOC1 expression were obtained in which expression levels were normalized relative to a reference gene (Liu et al., 2008; Gunl et al., 2009) (Figs. 2B–2C). In one of these (Fig. 2B), there was again a clear linear relationships between the observed SOC1 expression levels and flowering time in various backgrounds, with Pearson R2 value of 0.76. In the third one, there was less evidence for a linear relationship, with Pearson R2 value of 0.46 (p-value 0.3). Remarkably, it can be observed in Fig. 2 that one of the two parameters in the linear equation is quite similar for each of the three datasets (78, 98 and 91, respectively). This observation is more generally true, and we will come back to it in the next section. Note that the fits in Figs. 2B and 2C are less robust than the one in Fig. 2A, but we discuss below how we can combine multiple datasets for one gene in a simultaneous fit.

Dependency of flowering time on floral pathway integrator gene expression levels

Datasets reporting gene expression levels for various floral pathway integrator genes in different genetic backgrounds, in combination with flowering time values in these genetic backgrounds, were obtained (Fig. 3; Table 1). We start by fitting multiple models for each gene (one per dataset). Because in some cases, the number of data points in a dataset is rather small, we subsequently fit one model per floral pathway integrator gene (see below).

Figure 3 Overview of data and analysis.

(A) Available flowering time measurements and expression levels of floral pathway integrator genes were obtained from literature for various genetic backgrounds. (B) Genes from different upstream signalling pathways which were mutated in these genetic backgrounds are indicated. We analyse the data by modelling how expression level changes in floral pathway integrator genes (caused by genetic variation in the upstream signalling pathways) lead to quantitative changes in flowering time. In a first step, several models were obtained for each of the floral pathway integrator genes. Subsequently, one final model was obtained for each of these genes.

Table 1 Datasets obtained from literaturea.

Gene/reference	Mutant genotypes	Wildtype genotype	Conditionsb	Flowering timec	
SOC1					
El-Din El-Assal et al. (2003)	cry2, FLC-Sf2, FRI-Sf2, cry2; FLC-Sf2, cry2; FRI-Sf2, cry2; FRI-Sf2; FLC-Sf2	Ler, Cvi	LD, SD; 25C; day 21	TL	
Gunl et al. (2009)	gi, 35S::GI, 35S::GI; gi	Col	LD; 22C; day 15	TL	
Liu et al. (2008)	35S::AGL24, 35S::SOC1, agl24	Col	SD; 22C; day 21	RL	
FT					
Mizoguchi et al. (2005)	gi; 35S::GI,lhy, lhy;cca1, 35S::GI; lhy, gi; lhy; cca1	Ler	SD; 22C; day 10	TL	
Li et al. (2008)	agl24, 35S::SVP, svp, soc1	Col, Ler, C24	LD, SD; GA; 22C; day 11	TL	
Zuo et al. (2011)	cry2, cyr2; spa1	Col, RLD	LD; day 14	RL	
Fornara et al. (2009)	Cdf1, cdf2, cdf3, cdf5	Col	LD, SD; day 10	RL	
Yang et al. (2012)	35S:JMJ18, jmj18, tissue specific JMJ18	Col	LD; 22/18C; day 11	TL	
Gunl et al. (2009)	gi, 35S::GI, 35S::GI; gi	Col	LD; 22C; day 15	TL	
Endo et al. (2007)	cry2, tissue specific CRY2	Col	LD; day 9	RL	
Nefissi et al. (2011)	elf3; elf3 enhancer and suppressor lines	Col, Ler	LL; day 14	RL	
Sawa & Kay (2011)	gi; 35S::gi; tissue specific GI	Col	LD, SD; 23/16C; day 10	TL	
Tseng et al. (2004)	gi, spy	Col, Ler	LD; 22C; day 14	TL	
Wu, Wang & Wu (2008)	lwd1; lwd2, lwd1; lwd2/LWD1	Col	LD, SD; day 18	RL	
FLC					
Yang et al. (2012)	35S:JMJ18, jmj18, tissue specific JMJ18	Col	LD; 22/18C; day 11	TL	
El-Din El-Assal et al. (2003)	cry2, FLC-Sf2, FRI-Sf2, cry2; FLC-Sf2, cry2; FRI-Sf2, cry2; FRI-Sf2; FLC-Sf2	Ler, Cvi	LD, SD; 25C; day 21	TL	
He et al. (2004)	nox1, nos1, NO-donor treatment	Col	LD; 22C; day 10	RL	
Niu et al. (2007)	prmt10, prmt5	Col	LD; day 11	TL	
Jiang et al. (2007)	ldl1, ldl2, ldl1/ldl2	Col	LD; day 10	TL	
Wang et al. (2012)	ugt87a2	Col	LD; 22C; day 21	RL	
SVP					
Nefissi et al. (2011)	elf3; elf3 enhancer and suppressor lines	Col, Ler	LL; day 14	RL	
Li et al. (2008)	ft	Col	LD; 22C; day 11	TL	
LFY					
He et al. (2004)	nox1, nos1	Col	LD; 22C; day 10	RL	
Wang et al. (2012)	ugt87a2	Col	LD; 22C; day 21	RL	
AGL24					
Yu et al. (2002)	AGL24-RNAi, 35S-AGL24	Col, Ler	LD; 23C; day 5	RL	
Li et al. (2008)	agl24-1, 35S::SVP, svp-41, soc1-2	Col, Ler, C24	LD, SD; GA; 22C; day 11	TL	
Notes.

a Flowering time and expression data for specific floral pathway integrator genes were obtained from literature. Table includes data for each floral pathway integrator gene in which genetic background and expression data was measured. Values obtained from fitting each dataset are presented in Fig. 2 and Figs. S1–S5, and raw data are available in Data S1. Results of fitting these data using a linear model are shown in Table 2 and Table S1.

b Experimental conditions: LD indicates long day, SD indicates short day, LL indicates continuous light, GA indicates gibberellin. Day indicates age of plant for which measurements were taken. If reported, temperature is indicated as well.

c Flowering time measurement: RL indicates number of rosette leaves, TL indicates total number of leaves.

Table 2 Linear dependencies of flowering time on expression levelsa.

Gene	Normalization (number of datasets)	Sensitivity	T0	
SOC1	Scaled (1×)	−0.74	78.3	
	Actin (1×)	−72	97.5	
	Tubulin (1×)	−478.9	90.8	
FT	Scaled (3×)	−0.30 (0.06)	38.5 (6.0)	
	Actin (2×)	−19.6 (9.95)	45.4 (11.2)	
	Tubulin (1×)	−11.5	29.9	
	IPP2 (4×)	−4.0 (1.1)	53.4 (15)	
	UBQ10 (3×)	−363 (451)	45.8 (24.0)	
FLC	Scaled (7×)	5.8 (7.1)	12.7 (5.1)	
	Actin (1×)	81.0	8.1	
SVP	Scaled (1×)	0.29	4	
	Tubulin (1×)	37.2	−12.5	
LFY	Scaled (3×)	−5.0 (1.5)	14.6 (2.7)	
AGL24	Scaled (3×)	−1.7 (1.8)	19.6 (2.1)	
Notes.

a Values for parameters in linear fit T = Sensitivity * Expression Level + T0 for data shown in Fig. 2 and Figs. S1–S5. Normalization method used in the different datasets is indicated (scaled means normalization by scaling with wildtype or maximum expression value). Different normalization renders values of Sensitivity incomparable, but should not affect comparisons between values of T0. Reported values are average (standard deviation) in case multiple datasets are available for the same normalization. Characteristics of individual datasets are reported in Table 1. Values for Sensitivity and T0 in individual datasets are reported in Table S1.

As presented above for SOC1, linear relationships were observed between flowering time and gene expression levels (Figs. S1–S5; Table S1). These can be described by the following equation: (1) T=Sensitivity∗Expression level+T0.

Here, T is the observed flowering time, and the coefficients Sensitivity and T0 are specific for each floral pathway integrator gene. This equation describes how the measured flowering time T in a given genetic background can be modelled as a linear function of the expression level of a floral pathway integrator gene. The parameter Sensitivity describes the slope, in other words, the sensitivity of flowering time to changes in expression of the flowering time integrator network gene. Parameter T0 describes the intercept with the line where ExpressionLevel equals zero. Because, as explained above, expression data can only be directly compared if the same normalization has been applied, we present values of Sensitivity and T0 for each floral pathway integrator gene separately for each possible type of normalization (Table 2; Table S1). Figure S6 presents a histogram of the Pearson R2 values obtained with the different models, indicating that in the large majority of cases the value of R2 is higher than 0.75, meaning that more than 75% of the variation is explained by a simple linear model. The majority of the linear models has a significant p-value and this mainly depends on the number of datapoints available; for the cases with more than five datapoints, nine out of 12 have a p-value below 0.05 (Table S1).

In contrast to Sensitivity, T0 should not depend on normalization applied to the expression data (see ‘Methods’ for explanation). Hence, T0 values for the same floral pathway integrator gene obtained from different datasets should be quite similar. This was indeed observed for the SOC1 datasets presented above. More generally, although there is some variation, the different values of T0 obtained for a given gene are indeed significantly similar to each other compared to the values for the other genes (Text S1; Fig. S7). For the values of Sensitivity this is not the case, in line with our expectation.

One concern with respect to the analysis so far could be that for some of the datasets, the number of data points is rather small. We still chose to analyze such datasets initially separately because the combination of perturbations of various input pathways for the same floral pathway integrator gene allowed to demonstrate the similarity of T0 values. To further deal with the concern that some of the datasets are small, we subsequently fitted one final model per floral pathway integrator gene. This was done by allowing one T0 value per floral pathway integrator gene, but a different value of Sensitivity per dataset. In this setup, the number of data points is for each gene larger than the number of parameters; the number of degrees of freedom ranges from 2 for SVP to 72 for FT, and for all genes except SVP and LFY it is at least 30. Comparing the linear model predictions with the experimental flowering time values indicates in most cases a clear correspondence (Fig. 4A; Fig. S8). Note that FT has the most deviating behaviour in the sense that the relationship between experimental and predicted flowering time values is less linear.

Figure 4 Comparison between predictions and experimental data.

(A) Comparison between predicted and experimental flowering time for single linear model fitted to various SOC1 datasets. These datasets are the same as the ones used in Fig. 2, but here they are all fitted simultaneously using different values of Sensitivity but one single value of T0. The number of degrees of freedom in this fit is 30. (B) Comparison between T0 and flowering time of knock-out mutants. Based on fits of quantitative relationships between expression levels and flowering time, T0 predicts flowering time in knock-out mutants for different floral pathway integrator genes. These predictions show a good relationship with experimentally observed flowering time for these knock-outs. Each point in this plot represents one particular floral pathway integrator gene; red outlier point indicates ft.

The values of T0 are ordered as follows: T0,SVP <T0,FLC ∼T0,AGL24 ∼T0,LFY <T0,FT <T0,SOC1. T0 indicates the flowering time predicted by the linear relation in case of zero gene expression, which should be later for a flowering activator than for a flowering repressor. Hence, one would expect activators to have higher values than repressors. This is indeed the case. Given that the values of T0 indicate the expected flowering time when the level of a specific floral pathway integrator gene is set to zero, the values of T0 can be used to predict the flowering time for knock-out mutants of each of the floral pathway integrator genes. To validate these predictions, we compare them with our recently obtained set of flowering times for knock-out mutants (Leal Valentim et al., 2015) (Fig. 4B). There is a good correspondence between predictions and experimental data, although FT deviates from this pattern (Pearson R2 including all cases is 0.38 between T0 and flowering time of knock-out mutants; excluding FT, the value of R2 is 0.96 and the p-value ∼0.02). Note that LFY is not included in this figure because a lfy mutant does not flower properly at all (Blazquez et al., 1997). The discordant behaviour of LFY cannot be predicted by the simple linear analysis presented here. We provide an alternative analysis of our flowering time ODE model for prediction of LFY mutant flowering time in Fig. S9. LFY expression was fixed at given levels and the resulting flowering time predicted by the ODE model was recorded. For values of LFY below ∼1nM, the model predicts that there is no flowering. This behaviour is in accordance with the known behaviour of the lfy null-mutant which was not used for training the model, providing additional independent validation for the model.

The value of the slope of the fitted line in Fig. 4B is much lower than 1. This line relates the value of T0, our prediction of flowering time, to the observed flowering time in knock-out mutants. One reason for this small slope could be the fact that knock-out mutants in general will not have exactly zero expression in planta, leading to a smaller effect on flowering time than predicted. Nevertheless, the clear relationship between predicted and experimental flowering time provides independent validation of the simple linear model fits from which the value of T0 was obtained. Note that the flowering time and expression data used to obtain these fits are from genetic backgrounds in which upstream signal components have been mutated. Hence, the input data are independent from the floral pathway integrator gene knock-out mutants from which flowering time data is used in Fig. 4 for validation.

Discussion

Input from the environment is transduced by signalling pathways and integrated by a small number of floral pathway integrator genes. The complexity of the signalling pathways and their connection with the floral pathway integrator genes is overwhelming. Hence, understanding the effect of genetic variation in signalling pathways on flowering time is a daunting task. Our analysis indicates that in spite of this complexity, the effect of differences in genetic background can be quantitatively understood by focussing on expression level changes of floral pathway integrator genes. Perturbations in upstream signalling pathways effect floral pathway integrator genes mostly in such a way that the effect on flowering time is linear in the change in gene expression level. The fact that a linear response is significant in most cases, and that this response is observed for different floral pathway integrator genes, suggests that it is an important aspect of the way in which plants adapt to their local environment. The measured expression level changes are often up to tenfold or higher (Fig. 2, Figs. S1–S5). Hence, the linearity is observed over a large range of expression values.

Our findings on the role of gene expression variation in transducing the effect of genetic background variation to flowering time can be compared with more general analyses focusing on understanding the effect of variation in genetic background on phenotypes. For example, it was found in C. elegans that the effect of genetic background on the severity of RNAi and mutant phenotypes could be predicted from variation in the expression level of the affected gene (Vu et al., 2015). Also, it has been observed that genetic variation associated with trait variation is likely to influence expression variation as well (Nicolae et al., 2010), suggesting that this expression variation is intermediate in establishing the link between change in genotype and change in phenotype. A recent method estimated genetically regulated gene expression and correlated these estimates with phenotype values to identify genes involved in causing the phenotype (Gamazon et al., 2015). In a broad perspective, our analysis demonstrates the possibility of analysing the dependence of quantitative traits on expression of key genes involved, which could be applied to a variety of plant traits.

Our findings are based on literature data obtained under various experimental conditions. For example, the day or the timepoint during the day used for measurement is different between different datasets. More generally, gene expression clearly might display different trends in different tissues or between different cell-types within a tissue. Using a single qPCR-based value to characterize the expression of a gene ignores these spatial aspects completely. Although this puts limit on the level of comparability between these data sets, our analysis shows that it is possible to integrate such data. One additional complicating factor is the fact that qPCR data are reported in various ways. For one parameter in our model we overcome this problem by comparing data normalized in the same way. For the other parameter, this is not needed because it is independent of normalization. Nevertheless, the use of multiple qPCR reference genes would be of great value, both for better comparability between studies and also to ensure accuracy of measurements (Remans et al., 2014).

In addition to different ways of reporting expression, also different ways of reporting flowering time are used. The data we used either reported the total number of leaves, or the number of rosette leaves. Days to flowering is not often reported but would be a useful addition, in particular since leaf number and days to flowering are not always congruent (Takahashi & Morikawa, 2014). A more systematic storage of qPCR data and of phenotypic measurements (Krajewski et al., 2015) such as flowering time would clearly also be helpful to enable large scale comparative analyses such as we present.

The linear model appeared to be successful, but less so for FT than for other genes: the value of T0 obtained for FT did not correlate well with the experimental flowering time of an ft mutant (Fig. 4), and when fitting the various datasets simultaneously for each gene, there was a less clear linear relationship between predicted and observed expression for FT compared to the other genes (Fig. S8). This might relate to the fact that in particular for FT, the mRNA levels measured by qPCR are only a weak proxy for the real amount of active component. This is because FT protein is transported from leaves to meristem before it may exert its effect on SOC1 and FT. Molecular aspects of this transport are not known in much detail yet, but one could imagine that there would be some kind of threshold above which not all FT is transported. If this would be the case, the predicted value of T0 in our analysis would be too low, as is indeed observed when the predicted values are compared with experimental flowering times for mutants (Fig. 4). A similar threshold behaviour seems to be present in Fig. S8 for FT. A more general scenario in which the response of flowering time to expression level of a particular floral pathway integrator gene would not necessarily be expected to be linear is if multiple floral pathway integrator genes are simultaneously effected by upstream changes. Yet another complicating factor is the fact that various floral pathway integrator genes regulate each other. This could lead to correlations in expression levels of various floral pathway integrator genes, which in turn might influence our analysis. If a gene which is directly influenced by an upstream pathway regulates another floral pathway integrator gene, both might in principle display a clear correlation between flowering time response and expression level.

In the literature, the quantitative, continuous nature of flowering time and its gradual response to changing input is often neglected when analysing the effect of variation on flowering time. In many cases, the measured response of flowering time to perturbations is reported just as leading to early or late flowering. Only a few studies analyse quantitative relationships between gene expression levels and flowering time. This includes a study in which AGL24 is shown to be a dosage-dependent mediator of flowering signals (Yu et al., 2002). FLC levels in Arabidopsis accessions are correlated to flowering times of these accessions (Lempe et al., 2005). For rice, there is one example of analysis of quantitative relationship between expression of an FT ortholog and flowering time (Takahashi et al., 2009). Our comprehensive quantitative analysis neatly fits with these previous findings and quantifies the dosage dependence of flowering time for various floral pathway integrator genes. It indicates that the effect size of genetic variation in input pathways on flowering time can be understood via expression changes of floral pathway integrator genes. This proportional response of flowering time to upstream changes enables a gradual adaptation to changing environmental factors such as temperature and light. The continuous nature of flowering time is therefore an essential aspect of the potential of plants to adapt to various environments.

Supplemental Information

Supplemental Information 1 Supporting Figures and Tables

Click here for additional data file.

Data S1 Various literature-derived datasets analysed in the manuscript

Each sheet in the file contains data for one floral pathway integrator gene. These datasets were obtained from literature; see Table 1 in main text for references. Each dataset presented in the file consists of pairs of gene expression - flowering time, measured in various genetic backgrounds.

Click here for additional data file.

Supplemental Information 2 Example R-code used for analysis

Click here for additional data file.

Additional Information and Declarations

Competing Interests

Author Contributions

Data Availability

The authors declare there are no competing interests.

Aalt D.J. van Dijk conceived and designed the experiments, performed the experiments, analyzed the data, contributed reagents/materials/analysis tools, wrote the paper, prepared figures and/or tables, reviewed drafts of the paper.

Jaap Molenaar conceived and designed the experiments, reviewed drafts of the paper.

The following information was supplied regarding data availability:

The raw data has been supplied as Supplementary File.

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
