# Peer review of "Floral pathway integrator gene expression mediates gradual transmission of environmental and endogenous cues to flowering time"

_PeerJ, doi:10.7717/peerj.3197_

## Round 0.1 · original submission · Minor Revisions

Two independent reviewers were very positive about the approach, the results and the overall presentation. However, they pointed a few minor issues that could easily be addressed in order to improve the clarity (discussion item and figure issue) and impact (availability of the data) of the manuscript.

Reviewer 1 ·

Basic reporting

This manuscript is based on a model of flowering time in Arabidopsis that the authors published recently (reference 18). Here they use an array of published expression data and flowering-time data to show that there is a direct correlation between the expression levels of key floral integrator genes and flowering time. Although in itself this is perhaps not surprising since many of these floral integrator genes act rather late in the flowering process, it is striking that considering the genetic complexity of flowering control the expression levels of a few genes are sufficient to predict the outcome.
The manuscript is clearly written and conforms to Peer J standards. The figures are necessary and well labelled.
I would like the raw data that was used for the analyses to be made available. Although they collected a lot of RT-PCR and flowering-time data. The sources of these data are listed in Table I. However, the authors have made a considerable effort to collect all of these data together and I think all of the collected raw data should be made available, perhaps as supplementary information.

Experimental design

The analysis seems robust, but I am not a mathematician or modeller.
The research question and data analysis are well described.

Validity of the findings

The conclusions and outcome of the study are clear. In the Discussion they clearly state the difficulties of incorporating data collected under different conditions at different times by different experimentalists. I think an additional difficulty is that many of these genes are expressed at different times and some of them are actually expressed in flowers, such as SOC1 and LEAFY. So perhaps in some experiments the increase in SOC1 and LFY mRNA levels is because the plant has started to form flowers; if this is the case then a correlation between this increase in expression and flowering would be trivial. Perhaps the authors should engage a bit more in the Discussion on the difficulties inherent in the spatial and temporal regulation of these genes, which is rather underestimated by relying entirely on RT-PCR data.

Additional comments

None

Reviewer 2 ·

Basic reporting

Aalt Van Dijk and Jaap Molenaar present a straightforward but interesting study on the relationship between floral pathway gene expression and flowering time in Arabidopsis thaliana. To this end, they use both simulations using a previously published modeling framework and fitting procedures on expression data and flowering times measured under various conditions or in different genetic backgrounds. They conclude that flowering time depends linearly on the expression levels of floral pathway integrator genes, and that this gradual, linear response of flowering time to upstream expression changes enables Arabidopsis to gradually adapt to changing environmental factors.
I have very little to remark on the study. The methodology appears to be sound, the results are interesting, both from a systems biology and an evolutionary perspective, and the text is no-nonsense and well-written. The study contains both simulation results on an earlier model and experimental data fits that corroborate the simulation results. Overall, this is a well-performed study that generates interesting insights into the quantitative aspects of flowering time control, which, if confirmed in crops, might be useful in a breeding context.

A few minor remarks:

- I had some trouble with the grey dotted lines in Figure 1, which the legend says ‘indicate five-fold expression range around the expression level at day 10’ . I assume that this is the expression level at day 10 in the unperturbed model. Is it correct that this interval is determined by taking the expression level at day 10 and then dividing/multiplying it by sqrt(5) ? It might be good to indicate the actual level at day 10 in a separate color.

- line 296: ‘import’ should be ‘important’.

Experimental design

no comment

Validity of the findings

no comment

Additional comments

no comment

---

## Round 0.2 · accepted · Accept

To my mind all remarks have been adequately addressed, so the manuscript is suitable for publication.